# Local Scale Exposure and Fate of Engineered Nanomaterials

**DOI:** 10.3390/toxics10070354

**Published:** 2022-06-29

**Authors:** Mikko Poikkimäki, Joris T. K. Quik, Arto Säämänen, Miikka Dal Maso

**Affiliations:** 1Occupational Safety, Finnish Institute of Occupational Health, Työterveyslaitos, FI-33032 Tampere, Finland; arto.saamanen@ttl.fi; 2Aerosol Physics Laboratory, Physics Unit, Tampere University, FI-33014 Tampere, Finland; miikka.dalmaso@tuni.fi; 3Centre for Sustainability, Environment and Health, National Institute for Public Health and the Environment (RIVM), 3720 BA Bilthoven, The Netherlands; joris.quik@rivm.nl

**Keywords:** manufactured nanomaterial, engineered nanoparticles, atmospheric release, airborne pollutant, dispersion modeling, near source exposure, environmental exposure assessment

## Abstract

Nanotechnology is a growing megatrend in industrial production and innovations. Many applications utilize engineered nanomaterials (ENMs) that are potentially released into the atmospheric environment, e.g., via direct stack emissions from production facilities. Limited information exists on adverse effects such ENM releases may have on human health and the environment. Previous exposure modeling approaches have focused on large regional compartments, into which the released ENMs are evenly mixed. However, due to the localization of the ENM release and removal processes, potentially higher airborne concentrations and deposition fluxes are obtained around the production facilities. Therefore, we compare the ENM concentrations from a dispersion model to those from the uniformly mixed compartment approach. For realistic release scenarios, we based the modeling on the case study measurement data from two TiO${}_{2}$ nanomaterial handling facilities. In addition, we calculated the distances, at which 50% of the ENMs are deposited, serving as a physically relevant metric to separate the local scale from the regional scale, thus indicating the size of the high exposure and risk region near the facility. As a result, we suggest a local scale compartment to be implemented in the multicompartment nanomaterial exposure models. We also present a computational tool for local exposure assessment that could be included to regulatory guidance and existing risk governance networks.

## 1. Introduction

A high number of studies can be found on atmospheric dispersion of pollutants and its computational modelling [1,2,3,4,5,6,7,8,9], but the approaches used in the studies have not always been incorporated in atmospheric fate modelling of engineered nanomaterials (ENMs) [10]. One rarely studied field is the atmospheric release of ENMs from a point source; for this, the most relevant scenarios are continuous release from an industrial production facility [11,12], or an accidental release of a large amount of nanomaterial [13]. From the safety point of view, the atmosphere acts as a direct ENM exposure route to humans and wildlife as well as an indirect route to soil and water organisms due to the transport and deposition of ENMs [14,15,16]. However, limited information exists on adverse effects such ENM releases may have on human health and the environment.

Nanomaterial exposure assessment approaches [17,18,19,20,21,22] that consider direct release of ENM to the environment via the atmosphere, incorporating the nanomaterial-specific aerosol processes from release to fate, are currently rare. One of the few existing methods, in which detailed nanospecific processes have been considered, is implemented with a compartmentalized approach, SimpleBox4nano(SB4N) [23]. In the method, the atmospheric route is treated as a single large compartment (approx. 500 km × 500 km × 1 km at default regional scale) into which the released ENMs are evenly mixed; in such a case, the removal processes acting on the ENM are assumed to occur evenly over the whole volume to be considered. As such, it is well suited for its purpose of estimating a regional background concentration as part of screening-level exposure assessment. However, as ENM releases and removal processes can be localized, it can be argued that a comprehensive risk assessment should include a more spatially oriented component to assess the concentrations and exposure in the local environment of the engineered aerosol source (local hot spots). For these reasons, it is reasonable to investigate the spatial scales on which a fully mixed compartment becomes a relevant assumption for screening level exposure assessment, and, whether an uniformly mixed compartment can resemble the background concentrations far away from an ENM source at which the aerosol has diluted significantly.

Despite the importance of local scale exposure assessment [10], no local scale atmospheric modeling studies have been performed to date for ENMs. Instead, a regional scale model LOTOS-EUROS is available and has been applied to Europe scale nanoparticle fate modeling [24]. In the model, a typical grid size is 0.5${}^{{}^{\circ}}$ (Lat.) × 0.25${}^{{}^{\circ}}$ (Long.), that is approximately 56 km × 28 km, but the model can be used for grid sizes down to 3 km. Recently, LOTOS-EUROS was compared to SB4N using a grid of approximately 7 km × 7 km [25]. The comparison showed that the regional predicted environmental concentration ($PEC_{reg}$) of ENM in air calculated with SB4N was of the same order of magnitude as the mean of the spatio-temporal concentration distribution gained from LOTOS-EUROS, meaning that SB4N is capable of predicting regional concentrations. However, the upper end of the distribution from LOTOS-EUROS showed an order of magnitude higher concentrations, which might result from the grid points that are close to the ENM sources. Therefore, a local PEC is required in addition to regional background PEC for a screening level exposure assessment.

For atmospheric pollutants in general, European chemical regulation (REACH) instructs to estimate local concentrations and exposure near atmospheric sources. The current regulatory tool, namely the European Union System for the Evaluation of Substances (EUSES) [26,27,28], uses 100 m as an examination distance for human exposure estimation. The area closer than 100 m represents an industrial site or a so-called “standard environment”. This environment is thought to illustrate the pollutant exposure to humans in terms of annual average or reasonable worst-case values. It can be seen as an “average distance between the emission source and the border of the industrial site” [27]. In addition, EUSES can be used for local environmental exposure assessment for chemicals for which it demonstrates calculations for pollutant deposition from air to ground. For the local deposition determination, EUSES uses a circle with radius of 1000 m representing the local agricultural area [27,28,29]. The relation of these distances to the actual near-source deposition has to our knowledge not been studied in the published literature. This rises a question whether they can be reasoned for local exposure assessment, and to what extent the variation in ENM properties and atmospheric conditions affect the actual deposition distance [30].

ENMs remain as single free airborne particles relatively briefly due to their high and size-dependent deposition efficiency to surfaces, and their aging processes (mainly coagulation and condensation growth) [10]. These processes change the removal rate and properties of ENMs, sometimes drastically as shown earlier for ambient nanoparticles [31,32]. This introduces a geographical gradient to the concentrations of the released and deposited material that can be assumed to be steeper than for non-deposition gaseous pollutants [33]. In addition, the size-dependence of the deposition rate couples the removal rate to the advection distance from the source, adding a nano-specific process that can be used in estimation of ENM transport and deposition distances. This work develops a robust computational tool for estimating the local predicted environmental concentrations (PEC${}_{local}$) and relevant deposition distances including the particle size dependence. The tool aims at improving local scale fate and transport modelling of ENMs by being especially useful for improvement of other tools that either do not include particle behavior or for developing local scale scenarios for screening level exposure assessment.

As a starting point, we chose to study the release of an ENM to the atmospheric compartment from a point source and compare it to the assumptions of a uniformly-mixed compartment approach. We study the spatial scale at which the potential differences could be considered, and their dependence on the ENM properties. This was done by developing and applying an atmospheric dispersion model (ADiDeNano) that takes into account nanoparticle dilution and deposition, which generally dominate over other atmospheric nanoparticle processes [34]. The model is able to calculate the $PEC_{air}$ resulting from an airborne emission of ENMs, and, by calculating the subsequent dispersion and dry deposition flux to ground, it can estimate the total deposition to a certain area. The total deposition stands as a base for estimation of the distances at which the ENM is deposited, e.g., to soil environment, thus showing a potential to study the scale at which local effects become significant. Within the scope of the study, we chose SB4N [23], as an existing reference point, and explored the possibility of including $PEC_{local}$ to it to compliment the regional and continental assessments already implemented in the tool.

## 2. Materials and Methods

### 2.1. Atmospheric Dispersion and Deposition Model

We built a computational tool (ADiDeNano) to estimate environmental concentrations and ENM deposition distances for which the theoretical assumptions and derivations are presented in the following.

#### 2.1.1. Gaussian Plume Formulation

The atmospheric dispersion of nanoparticles can be mathematically described by advection-diffusion equation [35], which describes the evolution of the total particle concentration as a function of spatial location and time. It can be solved analytically by making multiple assumptions [35]. A point like source is assumed to emit particles with continuous rate *E* (g/s) at height *H*. The wind is chosen to be along the *x*-axis with a constant wind speed *U*, which with other parameters are assumed independent of time yielding an equilibrium situation. Furthermore, the wind speed is assumed to be high compared to the diffusion in *x*-direction, the ground surface is assumed to be flat and approximated as a plane (z=0). The resulting solution is the ’classic’ Gaussian plume equation:(1)$$\begin{array}{cc} C_{cg}\left(x,y,z\right)= & \frac{E}{2\pi U\sigma_{y}\sigma_{z}}\exp\left(\frac{-y^{2}}{2{\sigma_{y}}^{2}}\right)\left[\exp\left(\frac{-{\left(z-H\right)}^{2}}{2{\sigma_{z}}^{2}}\right)+\exp\left(\frac{-{\left(z+H\right)}^{2}}{2{\sigma_{z}}^{2}}\right)\right], \end{array}$$
where $\sigma_{i}$ are dispersion parameters in crosswind (y) and vertical (z) directions. They describe the spatial deviation of the concentration distribution, and can be calculated by parametrizations based on, for example, Pasquill stability classes. The classes illustrate the atmospheric stability from extremely unstable (a) to unstable (b), slightly unstable (c), neutral (d), slightly stable (e) and stable (f). While this is a simple parametrization based on the wind speed, sun elevation and cloud cover, there exists other more sophisticated ways to describe atmospheric mixing based on Richardson number or Monin-Obukhov theory [5]. However, these approaches require information either on the thermal and mechanical turbulence or net radiation and temperature gradients, which are more difficult to measure compared to the needs of Pasquill’s classification. In this study, we use the parametrizations of Klug [36] and Davidson [37] for the Pasquill stability classes. For Klug’s parametrization, the values for $\sigma_{i}$ can be defined by equations:(2)$$\begin{array}{ccc} \sigma_{y} & = & R_{y}x^{r_{y}},\\ \sigma_{z} & = & R_{z}x^{r_{z}}, \end{array}$$
where *x* is the downwind distance from the source in meters and the coefficients $R_{i}$ and $r_{i}$ are defined experimentally, see Table 18.3 of [38] (p. 866). As for the Davidson’s parametrization, the values for $\sigma_{i}$ in meters are defined by equations:(3)$$\begin{array}{ccc} \sigma_{y} & = & a_{y}x^{b_{y}+c_{y}\ln x},\\ \sigma_{z} & = & a_{z}x^{b_{z}+c_{z}\ln x}, \end{array}$$
where *x* is the downwind distance from the source in km. The coefficients $a_{i}$, $b_{i}$ and $c_{i}$ have been derived from experimentally verified Pasquill-Gifford curves [37]. In our experience, for unstable atmosphere (classes a, b, c), the vertical mixing $\sigma_{z}$ can have unrealistically high values causing concentrations to drop extremely fast as a function of downwind distance. Therefore, as also discussed by Davidson, the vertical mixing ($\sigma_{z}$) is limited to 5000 m, which is also a general practice in, e.g., EPA models. Appendix A presents the horizontal and vertical dispersion parameters for different stability classes used in this study.

#### 2.1.2. Nanomaterial Deposition Calculation

The classic Gaussian equation is valid for non-depositing gaseous compounds, and simulations without deposition have shown to overpredict measured concentrations in case of a particle phase depositing pollutant [39], such as the ENMs of interest in this study. To take the deposition ($v_{dep}$) and settling ($v_{set}$) velocities of particles into account, Ermak [40] derived an analytical solution for the Gaussian dispersion (Equation (Equation 16)). Later, Rao [41] showed that it can be simplified to a form
(4)$$\begin{array}{cc} C_{er}\left(x,y,z\right)= & \frac{E}{2\pi U\sigma_{y}\sigma_{z}}\exp\left(\frac{-y^{2}}{2{\sigma_{y}}^{2}}\right)\exp\left(\frac{-v_{set}\left(z-H\right)}{2K_{z}}-\frac{v_{set}^{2}{\sigma_{z}}^{2}}{8K_{z}^{2}}\right)\\ \; & \times [\exp\left(\frac{-{\left(z-H\right)}^{2}}{2{\sigma_{z}}^{2}}\right)+\exp\left(\frac{-{\left(z+H\right)}^{2}}{2{\sigma_{z}}^{2}}\right)\\ \; & \times \left(1-\frac{\sqrt{2\pi }v_{0}\sigma_{z}}{K_{z}}\exp\xi^{2}\mathrm{erfc}\xi \right)], \end{array}$$
where $K_{z}=U\sigma_{z}^{2}/2x$ is the turbulent diffusion coefficient in *z*-direction and by definition $\xi =\frac{z+H}{\sqrt{2}\sigma_{z}}+\frac{v_{0}\sigma_{z}}{\sqrt{2}K_{z}}$ and $v_{0}\;=\;v_{dep}\;-\;0.5v_{set}$. By assuming $v_{dep}=v_{set}=0$, the equation converges to the classic Gaussian solution. The use of the Ermak’s solution (Equation (Equation 4)) requires information on the particle settling and deposition velocities. The particle settling, due to gravitation, can be calculated from
(5)$$v_{set}=\frac{g\rho C_{c}d_{p}^{2}}{18\eta },$$
where *g* is the gravitational acceleration, ρ particle density, $C_{c}$ Cunningham slip correction factor [42] and η dynamic viscosity of air. Likewise, the particle deposition velocity $v_{dep}$ can be calculated from equation
(6)$$v_{dep}=v_{set}+\frac{1}{r_{a}+r_{c}},$$
where $r_{a}$ is the aerodynamic and $r_{c}$ canopy resistance [38]. The resistances $r_{a}$ and $r_{c}$ can be calculated from a semi-empirical model, such as, the one by Rannik et al. [43]. As we use a deposition model based on nanoparticle measurements in a pine forest [43], we have to assume that the persistence and deposition of ENMs are presumably comparable to naturally occurring nanoparticles [44]. Appendix A presents the settling and deposition velocities for nanoparticles calculated by the parametrization of Rannik et al. By using the deposition velocity and the ground level concentration (*z* = 2 m), we can calculate the deposition flux of nanoparticles to the ground as
(7)$$F\left(x,y,2\right)=-v_{dep}\cdot C_{tot}\left(x,y,2\right).$$

The equation describes the mass of particles deposited to ground per surface area in a certain time (g/m${}^{2}$s). With this formulation, we calculate the net nanomaterial deposition rate (g/s) at a certain downwind distance *x*:(8)$$R_{dep}\left(x\right)=-\int_{0}^{x}dx\int_{-\infty }^{\infty }F\left(x,y,2\right)dy,$$
which corresponds to the amount of nanomaterial deposited between the source and distance *x*. In fact, the fraction of the total particle mass deposited on the ground can be defined as $f_{dep}\left(x\right)=R_{dep}\left(x\right)/E$. Furthermore, the fraction of particles suspended in air $f_{susp}\left(x\right)$ is calculated from the horizontal particle flux in wind direction:(9)$$f_{susp}\left(x\right)=\frac{1}{E}\int_{0}^{\infty }\;dz\int_{-\infty }^{\infty }\;U\cdot \;C\left(x,y,z^{\prime }\right)dy.$$

The net deposition and suspension offer an effective way to determine the nanomaterial deposition over a large area and find out how far from the source the emitted material is deposited.

#### 2.1.3. Boundary Layer Reflection

The previous formulations for the airborne concentration are valid for an unbounded atmosphere. However, an infinite atmosphere is not always a physically realistic assumption. The pollutant dispersion is often limited by an inversion or boundary layer [39] for which different theoretical formulations exists. Yamartino derived a Gaussian plume equation for reflecting ground (*z* = 0) and planetary boundary layer (PBL, *z* = $H_{pbl}$) by adding reflection terms to the vertical dispersion [45]. However, Yamartino’s formulation (Equation (Equation 17)) does not take atmospheric particle deposition into account. Later, Rao derived an equation including both deposition and PBL reflection [41]. In his formulation, Equation (Equation 4) is extended by replacing *H* with $H_{1}=H+2jH_{pbl}$, and summing *j* over from −∞ to *∞*. This can be seen as adding artificial particle sources at effective source heights $H_{1}$.

However, for values *j* < 0, Rao’s formulation shows physically questionable behavior: the reflections from the boundary layer are affected by deposition, while the reflections from the ground neglect deposition, although they should be the opposite. We corrected this behavior by using absolute value $|H_{1}|$ to obtain the following equation for the concentration
(10)$$\begin{array}{cc} C_{ra}\left(x,y,z\right)= & \frac{E}{2\pi U\sigma_{y}\sigma_{z}}\exp\left(\frac{-y^{2}}{2{\sigma_{y}}^{2}}\right)\\ \; & \times \sum_{j=-\infty }^{\infty }[\exp\left(\frac{-v_{set}\left(z-\right|H+2jH_{pbl}\left|\right)}{2K_{z}}-\frac{v_{set}^{2}{\sigma_{z}}^{2}}{8K_{z}^{2}}\right)\\ \; & \times [\exp\left(\frac{-(z-|H+2jH_{pbl}{\left|\right)}^{2}}{2{\sigma_{z}}^{2}}\right)\\ \; & +\exp\left(\frac{-(z+|H+2jH_{pbl}{\left|\right)}^{2}}{2{\sigma_{z}}^{2}}\right)\\ \; & \times \left(1-\frac{\sqrt{2\pi }v_{0}\sigma_{z}}{K_{z}}\exp\xi_{1}^{2}\mathrm{erfc}\xi_{1}\right)]], \end{array}$$
where $\xi_{1}=\frac{z+|H+2jH_{pbl}|}{\sqrt{2}\sigma_{z}}+\frac{v_{0}\sigma_{z}}{\sqrt{2}K_{z}}$. Here, the j=0 term corresponds to the Gaussian dispersion with a single reflection from the ground (Equation (Equation 4)), whereas the j≠0 terms are the higher order reflections each with a different effective source height $H_{1}$. The solution has shown to rapidly converge to a limit as j→±∞. In fact, a number of j=±10 reflections yields to a tolerance of one percent [41]. In addition, one can note that by fixing the settling and deposition velocities to zero, Rao’s solution converges to the one of Yamartino.

#### 2.1.4. Mass Balance Correction

The solution by Rao (Equation (Equation 10)) has shown to overpredict concentrations near the source due to non-conserving mass [46]. In comparisons to experimental data sets by Doran and Horst [47], they found an overestimation of the ground level concentration with mean bias and mean absolute error of circa 21 percent. As a matter of fact, by forcing a mass balance to the concentration calculation, the prediction improves and has shown 7 percent absolute error when compared to experimental data [47]. The mass balance states that the sum of the vertical particle flux due to deposition and horizontal flux due to wind equals the source strength [48], shortly also as $f_{dep}\left(x\right)+f_{susp}\left(x\right)=1$ for each *x*. Thus, the concentration $C_{m}\left(x,y,z\right)$ from Equation (Equation 10) can be corrected as:(11)$$C_{m+1}\left(x,y,z\right)=\frac{C_{m}\left(x,y,z\right)}{f_{dep}\left(x\right)+f_{susp}\left(x\right)}$$
by iterating for *m* until a tolerance of $\left(1-f_{dep}\left(x\right)-f_{susp}\left(x\right)\right)$ < 1% is reached for each *x*.

### 2.2. Division between Local and Regional Environments

Typically, the Gaussian formulations are used to obtain concentration fields around the source. However, from these concentration fields, we can also obtain the position-dependent ground deposition rate, which for the case of environmental exposure assessment is of high interest. By integrating the deposition over the distance, we can compute the cumulative deposited mass of the emitted nanoparticle within a distance *x* from the source. Dividing this with the emission rate, we obtain the fraction of emitted material that is deposited within this distance, $f_{dep}\left(x\right)$.

Here, we have chosen the distance at which half of the emitted mass has been deposited, that is $f_{dep}\left(x_{50}\right)$ = 0.5, as a representative distance for assessing the spatial dispersion and deposition of a nanoparticle emission. Such a single parameter is useful in effectively illustrating the effect of emission and atmospheric properties, such as particle size, atmospheric stability or wind speed, have on the environmental exposure around the source. In the following sections, we show these dependencies for a number of emission properties and case studies.

We also note that in addition to being an illustrative parameter, $x_{50}$ presents a way to easily implement a nearfield-farfield approach to the atmospheric dispersion. Using the approach borrowed from occupational exposure assessment [49], we can divide the regional compartment into a near field (NF) and a far field (FF). $x_{50}$ is the parameter describing the distance to the NF/FF boundary, and can be seen as the size of the local compartment near an emission source. It can be shown (see Appendix B) that concentrations in the NF and FF can be calculated by solely using the sizes of the local ($x_{50}$) and regional ($x_{tot}$) compartments along with the emission rate *E* and the deposition velocity $v_{dep}$. This presents a computationally light method that allows implementing emission transformation processes but also accounts for the emission-dependent elevated concentrations near the sources.
(12)$$C_{N}=\frac{E}{2\pi x_{50}^{2}v_{dep}},$$
(13)$$C_{F}=\frac{E}{2\pi \left(x_{tot}^{2}-x_{50}^{2}\right)v_{dep}},$$

### 2.3. Simulation of the Maximal Range of $x_{50}$ Distances

To assess the effects the atmospheric and particle properties have on the $x_{50}$ distance, we ran the dispersion model for 1296 separate combinations of input values for ENMs, presented in Table 1, and for 486 combinations for a generic airborne pollutant. The values in Table 1 represent the maximal range of physically relevant values for each parameter.

For the generic pollutant, we used deposition velocities reported by Petroff and Zhang [50]. The deposition velocity of 0.01 cm/s corresponds to deposition of particle sizes roughly 0.1 to 1 μm to water, snow or ice surfaces, while the velocity of 0.1 cm/s resembles similar deposition of 1 to 4 μm particles on water or 60 to 400 nm particles on forest surfaces. The highest deposition velocity of 1 cm/s, used in this study, can be attained for particle sizes 3–15 μm (forest, water surfaces) assuming a particle density of 1500 kg/m${}^{3}$[50]. Moreover, for the ENM deposition, we used the parametrization of Rannik et al. for particle sizes from 10 to 500 nm [43], see Appendix A. The parametrization uses the particle effective density for which we used values from 1000 to 18,000 kg/m${}^{3}$ for engineered nanoparticles, highest values found for gold nanospheres [51,52].

### 2.4. Case Study Simulations of TiO${}_{2}$ Environmental Releases

For realistic environmental ENM concentration estimations, we modeled two atmospheric releases of TiO${}_{2}$ reported in the literature. Fonseca et al. measured a TiO${}_{2}$ release from a paint factory [11]. They estimated a TiO${}_{2}$ mass emission, via a stack of height 7.8 m, amounting 0.51 g/h (0.14 mg/s) during a powder pouring activity, and a yearly total emission of 900 g (29 μg/s) assuming a yearly consumption of 500 tons. Furthermore, Koivisto et al. measured nano-TiO${}_{2}$ emissions, originating from a spray coating process, via local exhaust ventilation to atmosphere [12]. They estimated an emission rate of 20 mg/min (0.33 mg/s) during the spray process and a yearly total emission of 110 g (3.5 μg/s) assuming 10 kg use of nano-TiO${}_{2}$.

For both case study simulations, we utilized realistic atmospheric conditions. We used a mean surface air temperature of 15 ${}^{{}^{\circ}}$C [53], a wind speed of 2.5 m/s [54], and an average planetary boundary layer height of 1 km [55]. Moreover, the atmospheric stability is commonly ranging from unstable to neutral (classes a–d) during daytime and neutral to stable (classes d–f) during nighttime. To estimate a yearly variation, we employed literature data measured in six locations [56], see Table 2. Assuming the day and night to be of equal length, we arrived at combined frequencies of occurrence P(a−f), which we use for the case study modeling of the yearly average deposition flux
(14)$$F_{avg}=P\left(a\right)F_{a}+\ldots +P\left(f\right)F_{f},$$
where the deposition fluxes $F_{i}$ are simulated for each stability class a–f separately. Based on the average deposition flux to ground $F_{avg}$, we calculated the predicted environmental concentrations in soil (g/kg) after *a* years of continuous emission from a TiO${}_{2}$ handling facility. We assumed a uniform mixing layer $H_{soil}$ of 0.05 m and a soil density $\rho_{soil}$ of 1500 kg/m${}^{3}$ [57].
(15)$$PEC_{soil}=F_{avg}\frac{a\cdot 365\cdot 24\cdot 60\cdot 60\;\sec}{H_{soil}\rho_{soil}}$$

## 3. Results

### 3.1. Ground Level Concentrations

We modeled the ground level (*z* = 2 m) concentrations for separate Gaussian formulations presented in Section 2.1.1–Section 2.1.4. Appendix A presents a comparison of the formulations with and without atmospheric boundary layer reflection and deposition to the ground. It can be seen that all formulations produce similar concentrations at early distances from the source. However, at higher distances after reaching the boundary layer, the classic Gaussian equation and Ermak’s solution produce lower concentrations in absence of reflections. Furthermore, without deposition, the concentrations are higher by Yamartino’s solution, and therefore, Rao’s formulation gives a more realistic estimation. In this study, we will focus on the mass balance corrected concentration as the Rao’s solution leads to overestimations due to non-conserved mass, Section 2.1.4.

As can be observed from Figure 1, the ground level concentration alters for varying nanoparticle sizes, atmospheric stability classes and source heights. For a low source height, a stable atmosphere (class f) yields highest ground level nanoparticle concentrations, whereas, for a higher source height, it produces in general lower concentrations. In contrast, an unstable atmosphere (class a) will provide enough mixing for the ENM to reach ground level earlier yielding higher concentrations at early and lower concentrations at later distances. In addition, the nanoparticle size has an effect to the concentrations due to deposition. Particles with a larger deposition velocity (diameters of 10 and 500 nm) will deposit faster than particles with a lower deposition velocity (diameter 100 nm) yielding higher airborne concentrations for particles with low deposition.

For other atmospheric pollutants, we expect similar dependence on the atmospheric stability and the source height as was observed for nanomaterials. However, the atmospheric behaviour of other pollutants can differ from nanomaterials as their deposition velocity can be considerably larger. The effects of the atmospheric stability and the deposition velocity on a generic airborne pollutant can be observed from Appendix A. It shows lower concentrations for higher deposition velocities having the largest effect in a stable atmosphere.

### 3.2. Maximum Ground Level Concentrations and Associated Distances

To understand potential human exposure (public and occupational health), we calculated the maximum ground level concentrations (*z* = 2 m) as a worst case estimation of the exposure concentration. Additionally, we determined the distances from the emission source at which the maximum concentration is reached. For the generic pollutant and ENM simulations, the maximum concentrations ranged from 0.1 ng/m${}^{3}$ to 47 μg/m${}^{3}$, see Appendix A, while the maximum concentration distances ranged from 10 m to 17 km (Appendix A).

Appendix A shows that the source height stands as the main driver for the ground level concentration. Sources closer to the ground yield three to five orders of magnitude higher concentrations in comparison to emission sources at higher altitudes, since the aerosol is more diluted before reaching the ground.

In addition, the atmospheric stability and the wind speed affect the maximum concentrations. For a low source height, an unstable atmosphere with high mixing and high wind speed can produce two to three orders of magnitudes lower maximum concentration than stable atmosphere with a low wind speed. In contrast, for a higher altitude emission, an unstable atmosphere provides enough mixing for the pollutants to reach ground level, and thus, yields to higher maximum concentrations than a stable atmosphere. This is due to the pollutants better staying at the altitude of the emission, and therefore, producing low ground level concentrations. Surprisingly, the particle size and deposition velocity do not affect the maximum concentrations.

The maximum concentration distances are only affected by the atmospheric stability class and the source height (Appendix A). Higher atmospheric stability leads to larger distances at which the maximum concentration is reached. However, this is only true for source heights of 10 and 50 m, whereas the low source height of 2 m always produces maximum concentration at a distance of 10 m, which is the smallest simulated distance. In other words, the largest concentration is evidently reached right at the source. By increasing the source height from 2 to 10 m, the maximum concentration distance alters from 10 m to 0.1–1 km, and, by further increasing the source height to 50 m, the maximum concentration is reached at distances of 0.2–17 km depending on the atmospheric stability.

While the maximum ground level concentration prevails as the relevant metric for the human exposure, the rate of nanomaterial deposition to ground will evidently determine the environmental impacts.

### 3.3. Distance of 50% Deposited

To estimate the local area near the ENM source, we calculated the distances at which 50% of the ENM has been deposited, that is $x_{50}$, providing an insight for the environmental exposure assessment (See Section 2.2). Figure 2 presents the fraction of ENM cumulatively deposited before the distance *x*, calculated by Equation (Equation 8). It illustrates the $x_{50}$ distance (vertical lines) ranging from 400 m to 200 km. Lower source height *H* and higher deposition velocities (particle sizes 10 and 500 nm) produce lower $x_{50}$ distances, whereas higher source height and lower deposition velocity (particle size around 100 nm) allows the nanoparticles to travel further away from the source. Moreover, Appendix A shows that only large enough deposition velocities allow the particles to deposit close to the source.

Figure 3 and Appendix A demonstrate the maximal theoretical variation in $x_{50}$ distance to assess the effects of different variables. For ENM simulations, the $x_{50}$ ranged from 330 m to over 500 km, whereas, for the generic pollutant simulations, the lower range of $x_{50}$ distance was obtained already at 150 m.

Figure 3a shows that small $x_{50}$ distances, below 1 km, are achieved for small particle sizes (10 nm), in a stable atmosphere (class f) with a low wind speed (1 m/s). The $x_{50}$ distances 1–10 km are attained for particle sizes of 10 and 500 nm in neutral and stable atmosphere with a low wind speed. Distances below 100 km are mainly obtained only for lower wind speeds. The particles with low deposition (100 nm) seem to be mostly transported further than 100 km away from the source.

The boundary layer height ($H_{pbl}$) has an effect only in unstable atmosphere, in which the atmospheric mixing is sufficient to transport particles to high enough altitudes to reach the PBL. In those high mixing cases, higher boundary layer heights gives a rise to larger $x_{50}$ distances. Similar observations can be seen from Figure 3b. Decreasing stability in the atmosphere and increasing wind speeds cause the pollutants to be travelled further away. For example, for very unstable atmosphere (class a), the ENM is quickly mixed to the air masses and transported away from the source. Hence, the concentrations near ground are lower, deposition near source is lower and $x_{50}$ distance is larger. In contrary, for stable atmosphere (class f), the ENM mixes slowly and deposit close to the source, thus, resulting smaller $x_{50}$ distances.

Small pristine nanoparticles (10 nm) have a tendency to deposit closer to the source in comparison to larger agglomerated particles (100 nm). However, a possible aggregation to natural particles might yield even larger particle sizes (500 nm) with high deposition velocities, causing the particles to deposit closer to the source. This can be better seen from Appendix A, which clearly shows the effect of the deposition velocity on the $x_{50}$ distance. As the deposition velocity decreases by an order of magnitude, the $x_{50}$ distance (median) decreases also an order of magnitude. The particle size becomes more important for situations of low atmospheric mixing that allow particles to stay near ground level eventually leading to deposition.

From other parameters, only source height affects the $x_{50}$ distance: a source close to the ground level leans towards earlier deposition.

Note that the $x_{50}$ distances calculated here represent the maximal range of deviation for mostly extreme input values. Therefore, no strong statistical conclusions can be made of the most common $x_{50}$ distances. Instead, to have an estimation for realistic cases, we simulated case study data of ENM emissions using commonly prevailing atmospheric conditions.

### 3.4. Case Study Simulations of TiO${}_{2}$ Environmental Releases

To have a realistic estimation of the human and environmental exposure concentrations, we simulated two TiO${}_{2}$ environmental releases reported in the literature, see Section 2.4. The case of Koivisto et al. resulted in a maximum ground level concentration of 0.12 μg/m${}^{3}$ at a distance of 12 m for the yearly average emission, see Appendix A for detailed crosswind and vertical concentration profiles. For a larger temporary emission rate during the spray coating process, the maximum airborne concentration increases to 2–11 μg/m${}^{3}$ depending on the atmospheric stability.

The total deposition to the area within 500 km from the source was 20%, and thus, $x_{50}$ distance is much larger than 500 km meaning that most of the emitted ENMs are transported far away from the source. However, the high deposition rate per surface area might yield to the highest environmental concentrations close to the source. Figure 4a shows the $PEC_{soil}$ after 10, 20, 30 and 100 years of continuous emission from the ENM handling facility, which represent the long term environmental accumulation and exposure assuming a fully persistent ENM.

To have an idea of potential risks, the $PEC_{soil}$ can be compared to predicted no effect concentration for soil organisms $PNEC_{soil}$, which is 1000 μg of nano-TiO${}_{2}$ per 1 kg of soil [14]. The concentration in soil reaches 0.7 times no effect concentration after 30 years, and further 2.3 times PNEC after 100 years of continuous emission. Moreover, values of $PEC_{soil}$ / $PNEC_{soil}$ larger than unity are only reached in the vicinity of the source, that is closer than 50 m. Thus, it takes several decades to a century to reach concentrations that can potentially have environmental effects mainly at the close proximity of the handling facility. Note that potential ENM transformation and decay in soil [23] is not considered here, which might affect the concentrations over such broad timescales.

For the yearly average emission in the case study of Fonseca et al., we achieved a maximum ground level concentration $PEC_{air}$ of 0.07 μg/m${}^{3}$ at a distance of 10 m for unstable atmosphere (class a), while a neutral atmosphere results in $PEC_{air}$ of 0.03 μg/m${}^{3}$ at a distance of 100 m presenting much lower concentrations closer than that, see Appendix A. In fact, Fonseca et al. measured the ground level concentrations and deposition to ground near the TiO${}_{2}$ handling facility, and found concentrations below the detection limit of the instruments. However, our calculations suggest that their sampling locations, which were at the close proximity of the handling facility, may have been too close, thus not being able to catch the ENM deposited further away from the source, at roughly 100 m. Additionally, for a larger temporary emission rate during the powder pouring, the maximum airborne concentration increases to 0.12–0.36 μg/m${}^{3}$ depending on the atmospheric stability.

Figure 4b shows the simulated $PEC_{soil}$. After a century of emissions, $PEC_{soil}$ is only 0.6 times the $PNEC_{soil}$ for TiO${}_{2}$ meaning low risk for the soil organisms even after such a long emission time. The spatial distribution of ENM in soil (Appendix A) reveals the effect of a taller stack, in which case, the ENM spreads wider and further away than for a shorter stack (Appendix A). Taller stack results in lower concentrations close to the source, while the maximum concentration is reached further away.

Assuming similar emission rates for other ENMs, such as carbon nanotubes (CNT) with a much lower no effect concentration (176 μg/kg [14]), the PEC could exceed the PNEC even further away from the source (40–550 m). It can happen also at shorter emission times, that is, less than 10 years for the spray process (Figure 4a), while for a taller stack and a higher emission rate from powder pouring (Figure 4b), the PEC would remain below $PNEC_{soil}$(CNT) for three decades.

### 3.5. Comparison of Dispersion to Fully Mixed Compartments

We performed a comparison of PECs from the dispersion model (ADiDeNano) to the multimedia exposure and fate assessment model SB4N. As a case study exploring the differences from Gaussian approach to the fully mixed regional compartment in SB4N, we used Fonseca et al. case study data as a basis [11]. The Gaussian profile shows that the nanomaterial mass is distributed downwind with a maximum airborne concentration of 30 ng/m${}^{3}$ at 100 m, while regional $PEC_{air}$ from SB4N is 2 × 10${}^{-3}$ ng/m${}^{3}$. As expected, the Gaussian profile shows multiple orders of magnitude higher concentrations near the source, while reaching similar concentration as SB4N at 100 km. Further away at the border of the regional compartment (500 km), Gaussian profile predicts one order of magnitude lower concentration.

Simultaneously, for soil, the SB4N predicts a steady-state $PEC_{soil}$ of 1.1 ng/kg, that is similar as calculated by the dispersion model, assuming a uniform mixing layer, at a distance of 120 km from the source after a year of emission or at 400 km after 5 years (Figure 4b). These represent the scales at which the assumption of a fully mixed regional compartment becomes comparable to the Gaussian dispersion profile.

Moreover, as an example of adding a local compartment to multicompartment exposure assessment, we compared the Gaussian dispersion to two fully mixed compartments (Equations (Equation 12) and (Equation 13)). For a $x_{50}$ distance of 1.7 km, the local compartment has a concentration of 0.16 ng/m${}^{3}$, while the regional has a concentration of 1.8 × 10${}^{-6}$ ng/m${}^{3}$. Concurrently, for the same case, the dispersion model results in ground level concentrations from 2 μg/m${}^{3}$ at the source to 7 ng/m${}^{3}$ at $x_{50}$ distance, and to 6 × 10${}^{-5}$ ng/m${}^{3}$ at 500 km. Thus, by adding a local compartment to accompany the regional, the estimations by the uniformly mixed approach come closer to the ones by the dispersion model. In spite of that, the concentrations are an order of magnitude lower for the uniform compartments, which likely results from the differences in the vertical mixing approaches. The vertical mixing in the Gaussian approach remains low at close proximity to the source, and slowly increases towards larger distances, whereas, in the uniform mixing approach, the air is fully mixed. Hence, the uniform mixing causes lower ground level concentrations in comparison to Gaussian dispersion, and thus, induces lower deposition near the source resulting in a larger deposition distance.

To better estimate the ground level concentrations, and thus, the deposition to ground in the fully mixed compartments, one could add a separate horizontally spaced compartment near the ground level, since the emissions most commonly occur relatively near the ground. As an alternative, instead of adding extra ground level compartment, one might also consider limiting the height of the air compartment to lower than the boundary layer height, especially, in a case of stable atmosphere as simulated here. In contrary, in an unstable atmosphere, we would expect the ENM to be mixed all the way to the boundary layer, and hence the benefit received from such a limitation or extra compartment declines.

## 4. Discussion

### 4.1. Local Scale of Exposure Estimation

Deposition from air to ground exhibits an important environmental exposure route of nanomaterials. Here, we discuss the factors affecting the ENM deposition and possible metrics for separating the local from the regional scale exposure assessment.

Firstly, we argue that the distance at which half of the nanomaterial has deposited, $x_{50}$, acts as a possible boundary separating local from the regional scale. We computed the $x_{50}$ distance for a range of particle sizes and atmospheric conditions varying from 0.15 km to values larger than 500 km. We found that, in general, increasing wind speeds and increasing instability in the atmosphere cause the pollutants to travel long distances, since the ENM is being quickly mixed to the air masses allowing them to be transported away from the source.

Many simulated cases result in $x_{50}$ distances lower than 100 km. This suggests that the deposition can happen closer to the source in comparison to the uniform regional mixing (500 km). In other words, over 50 percent of the deposited ENM could be deposited in an area that is smaller than 4% of the total regional area. There exists also a number of situations, in which the $x_{50}$ distance can be as low as 1–10 km. This happens mostly in neutral or stable atmosphere with low wind speeds, and, for pollutants with a high deposition velocity.

As for nanoscale particles the deposition behavior changes strongly with the particle size: a 10 nm particle having ten-times higher deposition velocity than a 100 nm particle, the smallest pristine particles tend to deposit closer to the source than the agglomerated particles (100 nm), which better stick with the atmospheric air flows transporting to much larger distances. It has also been discussed before that after growing to larger sizes, over 100 nm in diameter, ENMs may remain in the atmosphere for long times [58], and hence, travel far away from the original emission sources. Our nanoparticle simulations show that the smallest particles (10 nm, $x_{50}$ = 0.3–300 km) indeed deposit closer to the source in comparison to larger agglomerated particles (100 nm, $x_{50}$ = 200–500 km). However, as the particles continue to grow to coarser agglomerates (>500 nm), e.g., aggregation with natural particles, the deposition velocity increases affecting the transport and causing them to deposit closer to the source. The particle size becomes even more important factor for situations of low atmospheric mixing that allows particles to stay near ground level for long times eventually leading to deposition.

Earlier, Rao [41] modeled the deposition of large particles with sizes of tens of micrometers ($v_{dep}$ = $v_{set}$ = 10 cm/s, *H* = 30 m, $H_{pbl}$ = 1000 m and *U* = 5 m/s). His results show that $x_{50}$ distance is reached approximately at 1.5 km for a stable (classes e and f) and at 5.3 km for a neutral atmosphere (d). For an unstable atmosphere, $x_{50}$ is larger than 20 km at which the fractions of deposited pollutant are 12, 20, and 47 percent for classes a, b and c. Our results for ENMs show, in general, larger deposition distances as nanoparticles possess an order of magnitude lower deposition velocities than micrometer scale particles. Moreover, Williams et al. [59] calculated average lifetimes for atmospheric nanoparticles (3–100 nm). Near ground level, the lifetimes range from 15 min to 1.5 d, which correspond to average travel distances of 0.9 to 130 km assuming a constant wind speed of 1 m/s. For a higher wind speed of 10 m/s, the travel distances can be as high as 9 to 1300 km. These estimations are well in line with the $x_{50}$ distances calculated in this study for ENMs, ranging from 150 m to over 500 km.

Apart from the $x_{50}$ deposition distance, we studied the maximum ground level concentrations as they correspond to the worst case human and environmental exposure. We calculated the distances at which the maximum concentrations are obtained: ranging from 10 m to 17 km for varying ENM properties and atmospheric conditions. Thus, in many cases, the worst case situation is observed at a distance much closer than $x_{50}$ emphasizing the need for considering also the maximum concentration distance as a metric for the local scale exposure assessment. In general, the maximum ground level concentration in air ($PEC_{air}$) relates directly, due to deposition, to the maximum environmental concentration, $PEC_{soil}$. Therefore, the distances of maximum concentration in air are the same for the maximum $PEC_{soil}$. Hence, the maximum airborne concentration is related to not only worst case human exposure, but also to the highest environmental exposure.

The local scale acts as an important factor for the ENM exposure assessment near the emission source. Both the $x_{50}$ deposition distance and maximum concentration distance provide useful information for the needs of human and environmental exposure assessment. Together they can be used to estimate the local scale of exposure near nanomaterial and other pollutant sources.

### 4.2. Implications for Nanomaterial Environmental Exposure and Fate Assessment

Several properties of the ENM have impacts on the dispersion and deposition of the material. This affects the fate of the released material into different environmental compartments, such as air, water, sediment and soil, especially near the assumed source. Existing multimedia fate and exposure assessment tools, such as SB4N [23], have considered the atmospheric mixing to be uniform over regional and continental scales. However, for the atmospheric exposure route, considering high exposure potential near the source, SB4N could be developed to consider the local exposure by implementing a local scale compartment representing the local or near-field exposure (NF), while the already implemented regional scale seems to adequately estimate the far-field exposure further away from the source (FF). For this, the $x_{50}$ distance can be used as a metric to estimate the size of the local compartment along with the transfer rate from the local (NF) compartment to the regional (FF) using Equation (Equation 31) or (Equation 37).

Our study suggests that the extent of local compartment varies from a few tens of meters to hundreds of kilometers. As discussed earlier, we can find nanoparticle and atmospheric properties for which there is a need for a more detailed local exposure assessment, especially, in the cases of stable atmosphere with low wind speeds or in the case of highly depositing pollutant. However, there exists also such cases that the fully mixed regional box is a valid assumption, that is $x_{50}$ distances equal or larger than 500 km, and tend to happen for unstable atmosphere, high wind speeds and poorly depositing pollutants. Therefore, we suggests that the size of the local scale in SB4N could be adjusted by the particle and atmospheric properties in question.

In the scope of environmental exposure assessment, the considered dispersion model (ADiDeNano) can be seen as a good approach to estimate local concentrations in air and related deposition to soil for a local scenario near a point source. It can improve the characterization of point sources in regional-type models, such as SB4N, as it can be used to estimate near-source deposition and transport fractions. On the other hand, it can be used as a data evaluation tool for linking observed airborne concentrations to the source. It could be included as a part of a fit for nano implementation of current guidance in REACH for calculating $PEC_{local}$ [28], pp. 111–114.

Due to their size-dependent behavior in the atmosphere, different sized ENMs can be transported to various distances from the original pollutant source. The developed ADiDeNano model is fitting to estimate such size-dependent transport and deposition distances and can bring insight to the environmental fate of ENMs emitted from atmospheric sources. Based on the modeling results, nanoscale materials tend to travel further away from the source than the ones in micrometer sizes. Although, in the cases of high atmospheric mixing, the smallest nanoparticles (<10 nm) can deposit close to the source due to Brownian diffusion, which leads to higher soil and water concentrations near source and affects ENM transfer fluxes between environmental compartments.

### 4.3. Applicability to Other Atmospheric Pollutants

In addition to ENMs, the modelling approach presented is also applicable to other forms of depositing atmospheric pollutants, such as nano- and microplastic particles in the environment [60]. Based on the simulations in this study, larger microplastic particles ($v_{dep}$ > 1 cm/s) have a potential to be deposited close to the source, while smaller nanoplastics could travel far away from the original sources. Given that nanoplastics are observed in Alps [61], far away from suspected sources, it is clear that further investigation on the near and far field deposition and transport of such pollutants is needed. Apart from nanoplastics, the dispersion model developed in this study could be applied also for, e.g., incidental heavy metal emissions to assess local atmospheric exposure [62], or for fertilizer or pesticide emissions [63]. For instance, this approach could be used to derive nanomaterial relevant scenarios for pesticide fate modeling, such as FOCUS [64].

### 4.4. Implications for Chemical Regulations

Current chemical regulation in Europe uses a distance of 100 m for the estimation of human exposure near atmospheric sources of pollutants [26,27,28]. This so-called ’standard environment’ is thought to illustrate the pollutant exposure in terms of annual average or reasonable worst-case values around the industrial facility. Our simulations show that the maximum airborne concentration, hence worst case situation, is often reached directly at the source, i.e., at a distance of 10 m for emission sources near ground level. For higher emission heights, the ground level ENM concentration maximum can occur as far as 17 km from the source (Appendix A), while concentrations closer to the source are lower. Thus, an assessment done only for a distance of 100 m could miss the concentration maximum appearing at a later distance, see Figure 1b. Although the 100 m standard environment currently used in European chemical regulation captures the maximum concentrations in most cases, the standard environment might miss the worst-case exposure concentration for the case of high altitude emission, that is happening more than 10 m above ground.

In addition, in the regulatory local environmental exposure assessment, a distance of 1 km represents the local agricultural area near a pollutant source [27]. For most of our simulated cases, the ENM deposited further away than 1 km from the source, thus, an exposure assessment done at 1 km provides mostly a sufficient worst-case estimation. However, our simulations showed a few cases leading to ENM deposition closer than 1 km. These cases appear specifically for stable atmosphere with low wind speeds. Additionally, a high enough particle deposition velocity is needed for a rapid deposition near the original emission source (<1 km) being true for small nanoparticles (10 nm) and large micrometer scale particles (>3–15 μm). Moreover, from the maximum exposure point of view, the emission sources near ground level (height < 10 m) yield to the highest exposure concentrations, and, mostly attained at distances much closer than 1 km.

All in all, since there is no single explicit distance for the worst-case, but it rather ranges from 10 m to 17 km depending on particle size, atmospheric stability, wind speed and emission height, we suggest that the exposure evaluation distance is determined individually for the situation in hand by taking the emission height into account. This would ensure correct worst-case estimation in all situations.

### 4.5. Atmospheric Release of TiO${}_{2}$ as a Case for ENM Exposure Assessment

We simulated two cases of TiO${}_{2}$ atmospheric release: one from a paint factory [11] and another from a spray coating process [12]. The modeling results can be used as an indicative risk flagging to estimate exposures in large timescales. Even though the airborne ENM concentrations were low and well below the predicted no effect concentrations [14,65] in a short timescale, the ENM has a potential to accumulate to the soil environment by continuous deposition reaching quantities that, in a period of several decades to a century, could have effects on the soil organisms (PEC > 1000 μg/kg) in the local scale near the factory (see Section 3.4). Even so, the timescale for potential effects to occur are considerable with current emission rates and TiO${}_{2}$ consumption amounts. By increasing the TiO${}_{2}$ consumption ten- or hundredfold, we expect local environmental effects to occur after a decade or even less than one year, respectively. And therefore, before profoundly increasing the ENM consumption, one should assess the need for better emission control strategies.

For the two simulated cases, the TiO${}_{2}$ deposits to soil areas up to 150 m with maximum concentrations observed at distances of 12 m and 100 m for the spray coating and paint factory cases, respectively. At the paint factory, the emission happens at 2.6 times higher altitude leading to a larger deposition distance. We consider these distances to specify the local region around the production facility being of interest for local environmental exposure assessment, and highlighting the potential concentration hot spots. As we saw, depending on the emission height, the maximum concentrations could be reached at locations not directly adjacent to the ENM production or handling facility, and therefore, relevant environmental measurements might need to be performed further away from the ENM source. Appendix A present examples of such distances.

### 4.6. Implications for Occupational Exposure Assessment

The maximum airborne concentrations at ground level present the worst case situation of occupational exposure within an industrial site. For a range of atmospheric and nanoparticle properties, the maximum concentrations were reached at distances of 10 m–17 km, which from a theoretical point of view present the scale at which the exposure might happen. The highest concentrations were obtained for emission sources near ground level that pose the highest exposure potential near industrial facilities, that is less than 1 km for sources not higher than 10 m from ground. For emissions taking place at higher altitudes, the ground level concentrations seem to stay relatively low.

Furthermore, studies on atmospheric TiO${}_{2}$ emissions present realistic cases for occupational exposure potential. At a paint factory in a worst case [11], workers have a potential to expose to a concentration of 0.36 μg/m${}^{3}$ (at 100 m) during powder pouring if working or otherwise staying outdoors. On the other hand, a higher exposure potential of 11 μg/m${}^{3}$ is reached in the vicinity of a spray coating plant (at 12 m) [12], due to the emission source being close to ground level. Although the TiO${}_{2}$ has been classified as suspected of causing cancer (category 2, through the inhalation route), the potential exposure concentrations in both cases are far below, e.g., the recommended exposure limit (REL) of 300 μg/m${}^{3}$ set by NIOSH for ultrafine TiO${}_{2}$ particles [66]. However, since the TiO${}_{2}$ is classified as suspected carcinogen it is recommended to keep the exposure as low as technically possible.

Overall, the dispersion modeling approach presented in this study (ADiDeNano) offers a possibility to estimate occupational exposure from a novel point of view. The method can be advantageous particularly for exposure assessment at large production plants that include outdoor processes and work operations in open air. Furthermore, within the factory, the pollutants could be transported from outdoor air to indoor office spaces via general ventilation [67,68], which presents another possible occupational exposure scenario. A further application stands at construction sites at which bed rock drilling and blasting can produce substantial amount of dust [69]. The dispersion model introduces a possibility to estimate needed safety distances and waiting times after drilling or other work operations.

### 4.7. Implications for Public Health

If residential or otherwise populated areas are located near a ENM handling facility or an equivalent chemical factory, people could be exposed in long term to pollutants emitted from there. In case of the ENM emissions considered in this study [11,12], the potential yearly average exposure concentrations are at maximum 0.03 to 0.12 μg/m${}^{3}$, which are low in comparison to common urban background concentrations [70]. Even, the highest concentration (11 μg/m${}^{3}$) occurring only momentarily while the spray process takes place stands below the American and European annual limit values as well as the WHO guideline (24-h mean) for outdoor particle pollution, PM${}_{2.5}$ [71]. And therefore, as concentrations would be mostly below, there seems to be of little risk to people living nearby. However, if the ENM potentially emitted has a potential to be toxic to humans at low concentrations, such as rigid multi-walled carbon nanotubes [72], particular attention is necessary to control the emissions.

### 4.8. Limitations and Further Studies

It has been discussed that the Gaussian approach is appropriate for local scale risk assessment with the possibility to determine long-term average loads to the environment [5]. However, for complex geometries found in urban environments, a computational fluid dynamics (CFD) modeling might be needed as dispersion models cannot represent the detailed wind field near buildings [5]. However, CFD models are often too computationally expensive to use in risk assessment purposes where simple modeling system and fast response time are prioritized. Moreover, Gaussian modeling has shown to produce poor results in low wind speed situations at which the turbulent diffusion becomes more significant [5]. These situations are also ofttimes most dangerous in real-life due to connection to stable atmosphere and low inversion levels that can result in high ground level concentrations [5]. Therefore, for such cases, more complex models can be used to give more detailed estimations and, hence, improve the overall accuracy of the exposure assessment at extremely small scale of tens of meters from the source. One could also think of incorporating aerosol dynamic and chemistry processing to local scale risk assessment of ENMs, such as done in the MAFOR model for other atmospheric pollutants [34] to study the effects of such processes have on the exposure.

It is to be noted that for the local screening level assessment discussed in this study, we assumed a single point source from where the total nanomaterial mass is emitted from. For such worst case PEC and ENM deposition estimations, there is no need to estimate the exact ENM spatial distribution in the regional scale. However, if one aims at realistic prediction of ENM spatial distribution [73], one needs to consider a number of sources with arbitrary spatial distribution in the regional scale. If such atmospheric emission data from individual sources becomes available, the multiple source situation could be taken into account by placing and modeling multiple local point sources in the bigger regional scale. Nevertheless, although modelling only a single point source, as done in this study, is not directly useful to estimate the exact spatial distribution of pollutants at the regional scale, our analysis provided unique information on the scale in which the local effects need to be considered for screening level environmental risk assessment.

The lack of data on nanomaterial dispersion in the atmosphere arises a need for experimental studies on atmospheric ENM dispersion. Such studies and data sets would enable the validation of this type of models. Furthermore, for the case study simulations of mean annual deposition resulting from two TiO${}_{2}$ releases, we assumed yearly average values for the nanomaterial emission rate as there was no information on the diurnal or annual variations. However, one could simulate a typical year instead including realistic daily and yearly variations. This presents an opportunity for future modeling studies when measurement data on such variations becomes available.

## 5. Conclusions

Atmospheric near-source plumes can act as a medium to transport engineered nanomaterials (ENMs) from production and handling facilities to populated areas, while also presenting an exposure route to local soil and water environments due to deposition. Based on earlier developments in the literature, we build a robust atmospheric dispersion model coupled with a nanoparticle deposition description (ADiDeNano), and used it for simulations of atmospheric dispersion of ENMs from a stationary release source. We calculated the distances at which half of the emitted ENM mass has been deposited, $x_{50}$, showing a potential candidate to separate the local from regional scale exposure assessment in screening level modeling. We compared the ADiDeNano model to an existing multicompartment nanomaterial exposure model, SimpleBox4nano, using data from real atmospheric emissions from a paint factory. We showed that the near source local exposure concentrations from ADiDeNano are several orders of magnitude larger than the uniformly mixed regional concentrations (SB4N), and further explored the possibility of adding a local compartment to accompany the regional exposure assessment already implemented in SB4N. This would enable the estimation of local concentration hot spots near ENM sources in the scope of screening level environmental risk assessment.

Currently, the local screening level environmental exposure assessment is performed by a separate model (EUSES/CHESAR) [27,74,75,76]. By adding a local scale to multimedia exposure assessment (SB4N), all compartments from local to regional to continental would be directly connected by a single model allowing preservation of mass between compartments. Therefore, and for the reasons discussed earlier in Section 3.5 and Section 4.2, we suggest a local scale compartment to be added to existing multicompartment environmental exposure and fate assessment models, such as SimpleBox4nano [23] and MendNano [77], to complete the environmental exposure assessment.

As per our simulations, an appropriate size for the local compartment can range from circa 100 m to 10 km depending on ENM properties and prevailing atmospheric conditions. The use of smaller local compartment sizes would evidently lead to more conservative estimations. Furthermore, the exposure estimation distances of 100 m (human) and 1 km (environment) used in regulatory local scale chemical risk assessment seem to be, based on our simulations, in most cases adequate for catching the worst-case exposure concentrations. However, we found a few cases for which the worst-case concentration is reached either at really close proximity of 10 m or as far as 17 km from the source depending mostly on the emission height. Thus, a case-by-case determination of the evaluation distance would ensure correct worst-case estimation in all situations.

The simulations on two atmospheric emission cases of TiO${}_{2}$ reported in the literature yielded a relatively low exposure potential from the perspectives of environmental, occupational and public health. The long term environmental accumulation shows potential local effects only after several decades to a century depicting a low risk with current production volumes. However, a new assessment would be needed before profoundly increasing the ENM consumption to, e.g., ten- to hundredfold.

Altogether, the study has a wide range of implications on environmental and occupational exposure assessment as well as chemical regulations. Apart from ENMs, the results and the developed modeling tool (ADiDeNano) can be applied to many other atmospheric pollutants as well, such as nano- and microplastic among others. ADiDeNano has a potential to be used in everyday exposure assessment in contrast to more complex tools needing specialist knowledge [78]. It could be included as a part of a fit for nano implementation of current guidance in REACH for calculating local exposures [28], and it could be implemented to existing risk governance frameworks for nanomaterials [79].

## Figures and Tables

**Figure 1 toxics-10-00354-f001:**
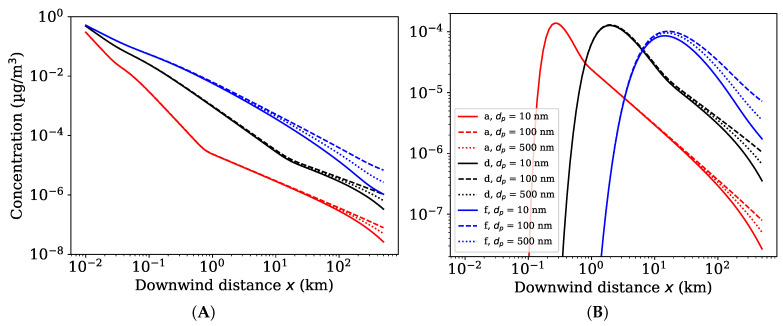
Ground level concentration (mass balance corrected) at the centerline of the plume as a function of downwind distance for a source height of (**A**) 2 m and (**B**) 50 m. Simulations for different particle sizes (10, 100 and 500 nm) and atmospheric stability classes (a, d, f) with a boundary layer height of 0.2 km and a wind speed of 10 m/s using Klug’s dispersion parametrization.

**Figure 2 toxics-10-00354-f002:**
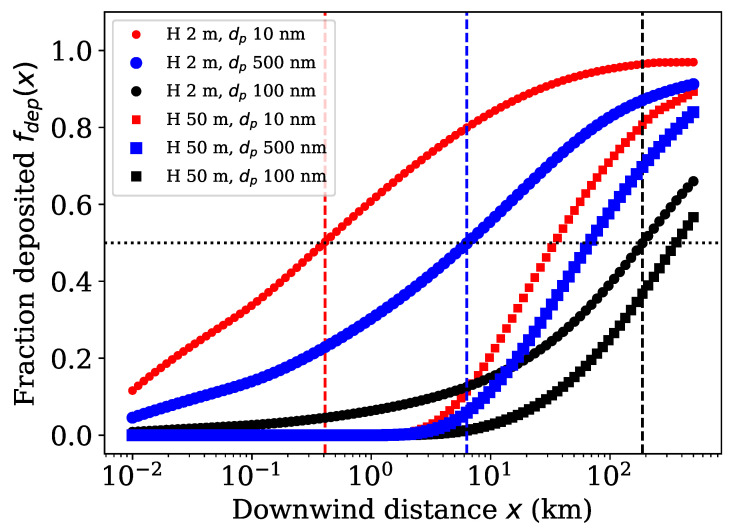
Fraction of ENM deposited to the ground for different particle sizes $d_{p}$ and source heights *H*. Vertical dashed lines represent the distances of 50% deposited, that is $x_{50}$.

**Figure 3 toxics-10-00354-f003:**
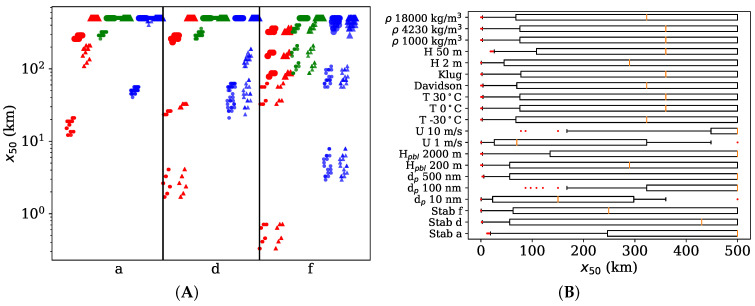
The $x_{50}$ distances for ENM simulations. (**A**) All data points grouped for the atmospheric stability classes (a, d, f). Circles correspond to the boundary layer height of 0.2 km and triangles the height of 2 km. Red markers have a particle size of 10 nm, green a size of 100 nm and blue a size of 500 nm. Smaller marker sizes represent the wind speed of 1 m/s and larger markers the speed of 10 m/s. (**B**) Sensitivity of the input parameters. The yellow line represents the median value of all observations for a certain parameter value. The boxes present 25 and 75% quartiles, while the whiskers correspond to the 5th and 95th percentiles. Red dots are outliers.

**Figure 4 toxics-10-00354-f004:**
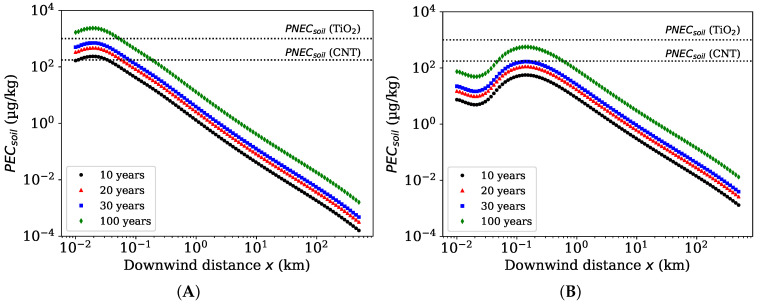
Modeled predicted environmental concentrations of nano-TiO${}_{2}$ in soil ($PEC_{soil}$) at different distances from the source after 10, 20, 30 and 100 years of continuous emission from (**A**) a spray coating facility Koivisto et al. and (**B**) a paint factory Fonseca et al., see Table 1. The horizontal lines represent predicted no effect concentrations in soil ($PNEC_{soil}$) for nano-TiO${}_{2}$ and carbon nanotubes (CNT).

**Table 1 toxics-10-00354-t001:** The ADiDeNano dispersion model parameter values for simulations of a generic pollutant, ENMs and case studies.

	Generic Pollutant	ENM	Fonseca et al. [11]	Koivisto et al. [12]
**Source properties**				
Emission rate *E* (μg/s)	29	29	29	3.5
Source height *H* (m)	2, 10, 50	2, 50	7.8 *	3 *
**Particle properties**				
Settling velocity $v_{set}$ (cm/s)	0.0001, 0.01, 0.1	Equation (Equation 5)	Equation (Equation 5)	Equation (Equation 5)
Deposition velocity $v_{dep}$ (cm/s)	0.01, 0.1, 1	Equation (Equation 6)	Equation (Equation 6)	Equation (Equation 6)
Particle diameter $d_{p}$ (nm)	-	10, 100, 500	260	280
Particle density ρ (kg/m${}^{3}$)	-	1000, 4230, 18,000	940	2100
**Atmospheric conditions**				
Air temperature *T* (K)	-	243.15, 273.15, 303.15	288.15	288.15
Wind speed *U* (m/s)	1, 2.5, 10	1, 10	2.5	2.5
Dispersion parametrization	Davidson, Klug	Davidson, Klug	Davidson	Davidson
Boundary layer height $H_{pbl}$ (km)	0.2, 1, 2	0.2, 2	1	1
Atmospheric stability class	a, d, f	a, d, f	d, a–f	d, a–f
Maximum distances *x*, *y* (km)	500	500	500	500

* Received from a personal connection with the TiO_2_ handling facilities.

**Table 2 toxics-10-00354-t002:** Average frequencies of occurrence P(i) of atmospheric stability class *i* in the Northern Hemisphere. The data is collected from [56].

Class	Daytime (%)	Nighttime (%)	Combined (%)
**a**	7	1	4
**b**	7	1	4
**c**	7	1	4
**d**	51	23	37
**e**	14	37	25.5
**f**	14	37	25.5

## Data Availability

The ADiDeNano (v1.0) dispersion model source code, developed and utilized for the simulations in this study, is available from https://doi.org/10.5281/zenodo.6700595 (accessed on 23 June 2022). The most current version and developments of the model are available at https://gitlab.com/MiPo/ADiDeNano (accessed on 23 June 2022). Table 1 displays the parameter values used in the simulations. The SimpleBox4nano model is available online at https://www.rivm.nl/en/soil-and-water/simplebox4nano (Version 4.01-nano 11.3.2021, accessed on 1 May 2022). Appendix A presents the applied parameter values.

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
