# Peer review of "Local Scale Exposure and Fate of Engineered Nanomaterials"

_toxics, 2022, doi:10.3390/toxics10070354_

Round 1
Reviewer 1 Report
The authors did a nice work. Minor drawbacks and recommended improvements.
1. Abstract:
The authors need to specify which nanomaterials the chosen cases are for.
2. Figure 4: Please define the meaning of the abbreviation PNECsoil (CNT) in the figure caption.
3. Further emphasis needs to be placed on the implications of the proposed model in terms of the environmental fate and behavior of nanomaterials.
4. Data Availability Statement:
The detailed information on the ADiDeNano (v1.0) is unavailable from the website provided. Please check it.
Author Response
Reviewer 1
The authors did a nice work. Minor drawbacks and recommended improvements.
Thank you for reviewing our work!
- Abstract: The authors need to specify which nanomaterials the chosen cases are for.
We chose TiO2 handling facilities as data for them was available in the literature. The information has been added to the abstract.
- Figure 4: Please define the meaning of the abbreviation PNECsoil (CNT) in the figure caption.
We added a sentence to the figure caption:
“The horizontal lines represent predicted no effect concentrations in soil (PNECsoil) for nano-TiO2 and carbon nanotubes (CNT).”
- Further emphasis needs to be placed on the implications of the proposed model in terms of the environmental fate and behavior of nanomaterials.
Thank you for the comment! The implications of our study to environmental exposure and fate on nanomaterials has been discussed in Chapter 4.2. The discussion of the proposed model is placed at the end of the chapter. We added some further discussion on the nanomaterial behavior based on our model:
“Due to their size-dependent behavior in the atmosphere, different sized ENMs can be transported to various distances from the original pollutant source. The developed ADiDeNano model is fitting to estimate such size-dependent transport and deposition distances and can bring insight to the environmental fate of ENMs emitted from atmospheric sources. Based on the modeling results, nanoscale materials tend to travel further away from the source than the ones in micrometer sizes. Although, in the cases of high atmospheric mixing, the smallest nanoparticles (<10 nm) can deposit close to the source due to Brownian diffusion, which leads to higher soil and water concentrations near source and affects ENM transfer fluxes between environmental compartments.
- Data Availability Statement: The detailed information on the ADiDeNano (v1.0) is unavailable from the website provided. Please check it.
Thank you for noticing! The Gitlab code repository has now been fixed and the model source code can be accessed. Additionally, we uploaded the v1.0 of the model source code to Zenodo with a permanent Doi.
Reviewer 2 Report
The paper deals with the air dispersion simulation of NMs. The subject is interesting and of increasing importance due to the continuous grow of NMs market. Moreover, it should be noted that, despite this market increase, there is a limited knowledge around the potential impacts of NMs in air and the corresponding health effects.
The captions in some of the supplementary figures should be checked.
In Section 2.4, the authors claim that, to calculate mean annual concentrations, they assume that atmospheric stability class is neutral. However, it would be better to simulate a whole typical day with a typical daily variation of atmospheric stability conditions than to hypothesize that the “mean” stability class is D, something that is not scientifically accurate. Moreover, in this case, they might use a realistic variation of the emission rate too, whereas when they carry out one calculation they do not take into account these changes.
In Section 3.1, lines 278-282, it is stated that “For low source heights, ………….distances”. Could the authors say what is happening with other pollutants?
Author Response
Reviewer 2
- The paper deals with the air dispersion simulation of NMs. The subject is interesting and of increasing importance due to the continuous grow of NMs market. Moreover, it should be noted that, despite this market increase, there is a limited knowledge around the potential impacts of NMs in air and the corresponding health effects.
Thank you for recognizing the importance of our work! We agree that there is a limited amount of information on the potential impacts on the environment and health effects. We edited a sentence in the abstract accordingly and added it also to the introduction:
“Limited information exists on adverse effects such ENM releases may have on human health and the environment.”
- The captions in some of the supplementary figures should be checked.
We checked the figure captions and updated them accordingly.
- In Section 2.4, the authors claim that, to calculate mean annual concentrations, they assume that atmospheric stability class is neutral. However, it would be better to simulate a whole typical day with a typical daily variation of atmospheric stability conditions than to hypothesize that the “mean” stability class is D, something that is not scientifically accurate. Moreover, in this case, they might use a realistic variation of the emission rate too, whereas when they carry out one calculation they do not take into account these changes.
Thank you for the critique! This is an extremely good research idea. We incorporated the yearly variation of the atmospheric stability to our dispersion modeling based on literature data (Kahl & Chapman, 2018, Atmos. Env. 187, 196–209, https://doi.org/10.1016/j.atmosenv.2018.05.058.).
We added a description of the method, which incorporates the variation of the stability class to the Section 2.4 of the manuscript:
“To estimate a yearly variation, we employed literature data measured in six locations, see Table 2. Assuming the day and night to be of equal length, we arrived at combined frequencies of occurrence P(a-f), which we use for the case study modeling of the yearly average deposition flux.
… Equation 14 …
where the deposition fluxes Fi are simulated for each stability class a-f separately. Based on the average deposition flux to ground Favg, we calculated the predicted environmental concentrations in soil (g/kg)
… Equation 15 …
after a years of continuous emission from a TiO2 handling facility. We assumed a uniform mixing layer of 0.05 m and a soil density of 1500 kg/m3.”
As our dispersion model assumes a time-independent steady-state concentration plume, we decided to average separate deposition fluxes Fi based on the frequencies of the stability classes. This was a way to include the variation in the atmospheric stability providing physically relevant results but keeping the model as simple as possible.
We updated Figures 4 and S13 as well as the numerical values reported in Sections 3.4 and 3.5 based on the simulations performed taking the variation of atmospheric stability into account. At the end, only minor differences were obtained, thus not affecting the conclusions of the manuscript.
In addition, we unfortunately have no information on the actual diurnal or yearly variation in the emission rate, thus a yearly average value was used as calculated by Koivisto et al. and Fonseca et al. in their studies for the total yearly usage of nanomaterials in the handling facilities. We added discussion to the Chapter 4.8 as a limitation of our study and an opportunity for future modeling studies:
“Furthermore, for the case study simulations of mean annual deposition resulting from two TiO2 releases, we assumed yearly average values for the nanomaterial emission rate as there was no information on the diurnal or annual variations. However, one could simulate a typical year instead including realistic daily and yearly variations. This presents an opportunity for future modeling studies when measurement data on such variations becomes available.”
- In Section 3.1, lines 278-282, it is stated that “For low source heights, ………….distances”. Could the authors say what is happening with other pollutants?
We edited the sentence in question to a form:
“For a low source height, a stable atmosphere (class f) yields highest ground level nanoparticle concentrations, whereas, for a higher source height, it produces in general lower concentrations.”
We also added discussion on other pollutants at the end of the Section 3.1:
“For other atmospheric pollutants, we expect similar dependence on the atmospheric stability and the source height as was observed for nanomaterials. However, the atmospheric behaviour of other pollutants can differ from nanomaterials as the deposition velocity can be considerably. The effects of the atmospheric stability and the deposition velocity on a generic airborne pollutant can be observed from Figure S4. It shows lower concentrations for higher deposition velocities having the largest effect in a stable atmosphere.”